# Prevalence and Risk Factors of Bone and Dental Lesions in Neotropical Deer

**DOI:** 10.3390/ani14131892

**Published:** 2024-06-27

**Authors:** Tamires Ataides Silva, Andressa da Silva Martins, Lisandra Rodrigues Alves, Luana Wenceslau Bittencourt Pereira, Júlia Rebecca Saraiva, José Maurício Barbanti Duarte, Eveline dos Santos Zanetti, Christiane Marie Schweitzer, Iveraldo Santos Dutra, Ana Carolina Borsanelli

**Affiliations:** 1Posgraduate Program in Animal Science, School of Veterinary Medicine and Animal Science, Universidade Federal de Goiás (UFG), Goiânia 74690-900, GO, Brazil; tamires.ataides@gmail.com (T.A.S.); lisandra.rodrigues@discente.ufg.br (L.R.A.); 2Department of Veterinary Medicine, School of Veterinary and Animal Science, Universidade Federal de Goiás (UFG), Goiânia 74690-900, GO, Brazil; zandressa@discente.ufg.br (A.d.S.M.); luanawenceslau@discente.ufg.br (L.W.B.P.); 3Department of Production and Animal Health, School of Veterinary Medicine of Araçatuba, São Paulo State University (Unesp), Araçatuba 16050-680, SP, Brazil; julia.rebecca@unesp.br (J.R.S.); iveraldo.dutra@unesp.br (I.S.D.); 4Deer Research and Conservation Center (NUPECCE), School of Agricultural and Veterinary Sciences (FCAV), São Paulo State University (UNESP), Jaboticabal 14884-900, SP, Brazil; mauricio.barbanti@unesp.br; 5Marsh Deer Conservation Center (CCCP), School of Agricultural and Veterinary Sciences (FCAV), São Paulo State University (UNESP), Jaboticabal 14884-900, SP, Brazil; eveline.zanetti@tijoa.com; 6Department of Mathematics, School of Engineering of Ilha Solteira, São Paulo State University (Unesp), Ilha Solteira 15385-000, SP, Brazil; christiane.schweitzer@gmail.com

**Keywords:** bone resorption, broken mouth, dental biofilm, periodontal lesions, periodontitis, tooth wear

## Abstract

**Simple Summary:**

The study investigated bone and dental lesions in 180 dry skulls from eleven neotropical deer species, originating from both captive and wild conditions, through direct visual inspection. The research aimed to characterize these lesions, estimate their prevalence, and assess potential risk factors. The results showed a high prevalence of bone and dental lesions in all analyzed species, with dental calculus being the most common alteration (96.7%), followed by dental wear (71.1%). A positive correlation was found between animal age and the presence of lesions, indicating that older animals exhibited more bone and dental alterations. The prevalence of these lesions was similar between sexes. However, lesions were more common in captive-bred animals, likely due to their older age and a less diverse diet. Specific species, such as *Blastocerus dichotomus* and *Mazama americana*, were most affected by bone resorption and dental trauma. The study concluded that all eleven evaluated species are susceptible to bone and dental lesions, highlighting the importance of monitoring oral health and diet in captivity as fundamental practices for the conservation of these deer species.

**Abstract:**

Bone and dental lesions have been documented in various deer species globally, affecting the efficiency of ingestion and digestion, consequently influencing their general health and leading to a decline in survival and reproductive performance. The present study aimed to characterize bone and dental lesions in the dry skulls of individual deer, estimate the prevalence of these lesions, and assess potential risk factors associated with the development of bone and dental alterations. This study assessed bone and dental lesions in 180 dry skulls of eleven neotropical deer species, originating from both captivity and wildlife conditions, through direct visual inspection. A high prevalence of bone and dental lesions was observed in all analyzed species. Dental calculus was the most common alteration (96.7%), followed by dental wear (71.1%). Animal age positively correlated with most bone and dental alterations, indicating that older animals showed more lesions. Additionally, the prevalence of these alterations was similar between sexes. Moreover, all lesions were more common in captive-bred animals, likely attributed to their older age and a less diverse diet. *Blastocerus dichotomus* and *Mazama americana* were most affected by bone resorption and dental trauma and had the highest dental calculus prevalence, along with *Subulo gouazoubira* and *Passalites nemorivagus*. All eleven species evaluated in the present study were susceptible to the occurrence of bone and dental lesions. Therefore, monitoring oral health and diet in captivity are fundamental practices for the conservation of these species.

## 1. Introduction

Deer belong to the order Cetartiodactyla and suborder Ruminantia [1]. The Cervidae family is the second most diverse in the order, with eight neotropical deer species described in Brazil, including *Mazama americana*, *Subulo gouazoubira*, *Blastocerus dichotomus*, *Passalites nemorivagus*, *Ozotoceros bezoarticus*, *Mazama nana*, *Mazama bororo*, and *Odocoileus virginianus* [2,3,4,5,6,7,8]. Deer, as ruminants, rely on their teeth for mastication and food digestibility, which directly influence their overall health. Mastication is crucial for reducing food to a size that allows effective ruminal microbial digestion of cellulose in plant walls [9,10].

Consequently, the general health of ruminants is directly related to digestive efficiency, and teeth and periodontium alterations can have significant adverse effects [10]. Studies evaluating in situ roe deer (*Capreolus capreolus*) and moose (*Alces alces*) oral health indicated that molar teeth deterioration was associated with a decline in survival and reproductive performance in aged animals [11,12,13]. Among various oral conditions, periodontal disease is a significant concern in veterinary medicine as it affects the periodontium and triggers an immunoinflammatory response, exacerbated by oral microbiota imbalance, leading to irreversible consequences and eventual tooth loss [14,15,16].

Oral disorders have been documented in several deer species worldwide, including *Rangifer tarandus*, *Cervus elaphus*, *Cervus nippon*, *Alces alces*, *Dama dama*, *Odocoileus virginianus,* and *Blastocerus* [17,18,19,20,21,22,23,24,25]. These studies highlighted a range of dental and skeletal alterations, such as dental abscesses and periodontal lesions, observed in different geographical regions like the Czech Republic, Brazil, and the United States [20,21,25]. These conditions significantly impact the oral and digestive health of cervids, underscoring the necessity for ongoing investigations to elucidate their etiology, potential risk factors, and implications for overall health and digestive capacity.

Research focused on oral alterations in neotropical deer species is particularly scarce in the literature. Given the classification of numerous cervid species as threatened or vulnerable, understanding the prevalence and effects of these conditions is crucial for their conservation [7]. Oral alterations can affect survival and reproduction in these species. Therefore, this study aimed to investigate the occurrence of bone and dental lesions in dry skulls of neotropical deer. The use of dry skulls allows for a comprehensive assessment of dental alterations across multiple individuals, representing different species, locations, and years, providing broader insights into associated risk factors. This methodological approach was chosen to address practical challenges in sampling live deer in the field, facilitating a more extensive study of dental conditions in these populations.

## 2. Materials and Methods

### 2.1. Sampling

A total of 180 dry skulls of 11 species of neotropical deer, consisting of bone remains optimally preserved for study, were examined: 47 (26.1%) specimens of *Mazama americana*, 43 (23.9%) of *Subulo gouazoubira*, 24 (13.3%) of *Blastocerus dichotomus*, 13 (7.2%) of *Passalites nemorivagus*, and 9 (5%) of *Ozotoceros bezoarticus*, 4 (2.2%) of *Mazama temama*, 4 (2.2%) of *Mazama nana*, 10 (5.6%) of *Mazama bororo*, 2 (1.1%) of *Mazama chunyi*, 1 (0.6%) of *Mazama pandora*, 1 (0.6%) of *Mazama bricenii*, and 12 (6.7%) specimens of *Mazama* genus were unable to identify the species, and in 10 (5.6%) specimens, the genus and species were not identified. The samples were obtained over a period from July to November 2022.

Of the 180 specimens evaluated, 47 (26.1%) were wildlife animals (in situ), 84 (46.7%) were captive-bred (ex situ), of which approximately 23 (27.4%) were donated by other wildlife enterprises, and 49 (27.2%) did not have identification regarding their origin or how long they remained under human care. The animals were also classified into three age categories: young (1–12 months), adults (1–10 years), and seniors (above 10 years), based on information available in the individual records of each animal. We evaluated tooth eruption to estimate the age of individuals, as many animals did not have this information recorded. Furthermore, in a complementary manner, for animals lacking data regarding age in years, dentition was used to estimate whether the animal was young (deciduous dentition) or adult (presence of all permanent teeth). 

The dry skulls were from the museum of the Cervid Research and Conservation Center Museum (NUPECCE) at the São Paulo State University (UNESP) in Jaboticabal, São Paulo and the Marsh Deer Conservation Center (CCCP), Jaboticabal, São Paulo. The collection consisted of animals captured in the wild and animals kept in captivity in the two conservation centers mentioned above. These specimens were sourced from wildlife rehabilitation centers, delivered by environmental authorities, or obtained from zoos, natural reserves, roadkill incidents, and other museums. They represent animals from various countries in South and Central America, including French Guiana, Mexico, Paraguay, Venezuela, Bolivia, Argentina, and Peru. Causes of death varied and included pneumonia, hemorrhagic disease, neoplasia, sepsis, capture myopathy, and cervical fracture. Notably, 11 animals had documented histories of dental problems during their lifetime. All available skulls from the museum collection were utilized in this study.

### 2.2. Direct Visual Inspection 

All dry skulls were visually examined and photographically documented. The results obtained were individually cataloged in dental files, adapted specifically for the studied species. In the bone and dental evaluations, we considered several parameters: dental calculus degree (grade 0: absence of dental calculus; grade 1: less than 25% of the tooth covered by calculus; grade 2: between 25 and 50% of the tooth covered by calculus; grade 3: more than 50% of the tooth covered by calculus), tooth wear degree (grade 0: absence of enamel wear; grade 1: discreet wear; grade 2: wear up to 1/3 of the crown length; grade 3: wear exceeding 1/3 of the crown), tooth loss, non-functional teeth, dental trauma, furcation exposure, fenestration presence or absence, bone bulging presence, bone resorption presence or absence, dehiscence presence or absence, and dentin exposure presence or absence [26,27]. 

Non-functional teeth were defined as teeth that lost their mastication utility and no longer perform their primary function due to dental support loss, evidenced by adjacent alveolar bone. However, this assessment was subjective, and no tooth viability test was used as they were skulls. Three types of bone resorption were considered in this analysis, referring to the distance between the cementoenamel junction and the alveolar crest on the buccal surface, fenestration, and dehiscence [27]. The classification of periodontal disease was limited to the alveolar bone evaluation due to the absence of soft tissues [27].

All information regarding the individuals was duly catalogued. The information was recorded according to sex, species, age, origin (wildlife or captivity), and cause of death and compiled into spreadsheets.

### 2.3. Feeding Management of Deer under Human Care 

The diets of the captive-bred animals consisted mainly of concentrated feed supplement formulated for horses, with the following composition provided by NUPECCE: moisture (max) 13%, crude protein (min) 10%, ether extract (min) 2%, fibrous matter (max) 17%, mineral matter (max) 20%, calcium (max) 2%, phosphorus (min) 0.6%, and digestible energy 2900 kcal/kg, including the addition of molasses for enhanced palatability. In addition, the diet included roughage composed of leaves of *Morus* sp. (blackberry), *Hibiscus rosa-sinensis* L. (hibiscus), *Neonotonia wightii* (perennial soybean), *Pennisetum purpureum* (napier grass), *Boehmeria nivea* L. (ramie), *Medicago sativa* (alfalfa), *Cenchrus purpureus* (elephant grass), and *Leucaena leucocephala* (leucena).

### 2.4. Statistical Analyses

Descriptive and comparative analyses of the data were performed using R software. Pearson’s correlation test was utilized to analyze correlations between the variables: age, tooth wear, fenestration, dehiscence, tooth loss, bulging, fracture, furcation exposure, and teeth without function, which presented values ranging from −1 to +1. If the value approached 1, it was a positive correlation (an increase in the value of one variable interfered with an increase in the value of another variable), and if the value approached −1, it was a negative correlation (an increase in the value of one variable interfered with a decrease in the value of another variable). Values around 0 do not have a linear correlation. The relationship between the variables of sex, animal origin (ex situ or in situ), age, and dental lesions was evaluated using the Chi-square test. If violations were found for the Chi-square test, alternative methods such as Fisher’s exact test were analyzed. The significance level for both tests was set at 5% (*p* < 0.05).

## 3. Results

### 3.1. Species

All species presented at least one specimen with dental calculus, tooth wear, and dentin exposure (Table 1). There was a significant difference between the presence of dental calculus and the species evaluated (*p* ≤ 0.05), with the most affected species being *B. dichotomus* (23 of 24), *S. gouazoubira* (all 43), *P. nemorivagus* (all 13), and *M. americana* (all 47). Likewise, the presence of tooth fractures showed significant variation between species (*p* ≤ 0.05), with *M. americana* (24 of 47) and *B. dichotomus* (19 of 24) having the highest prevalence. Bone resorption also revealed statistically significant differences between species (*p* ≤ 0.05), with *B. dichotomus* (19 of 24) and *M. americana* (24 of 47) standing out as the most affected. Only in the species *M. pandora* and *M. bricenii*, no tooth eruption was observed. There was no significant difference between the presence of tooth wear, dentin exposure, tooth loss, non-functional teeth, furcation exposure, fenestration, dehiscence, bone bulging, and the species evaluated.

### 3.2. Sex 

We observed that 85 (47.2%) evaluated animals were male, 94 (52.2%) were female, and 1 (0.6%) did not have any information about sex. Using the Chi-square test and the Fisher exact test, no dependence was shown with any variable by the sex variable.

### 3.3. Ages

Regarding age, 9 (5%) animals evaluated were young, 61 (33.9%) were adults, 16 (8.9%) were senior, and 94 (52.2%) specimens had no record or information available about these specific data. Regarding these animals without data, 78 (83%) were considered adults, and 16 (17%) were classified as young based on their dentition. Among the evaluated animals, 39 (21.7%) presented signs of tooth eruption, indicating that these were very young animals. The teeth in the stage of eruption with the highest occurrence were the right maxillary third molar, the right mandibular third molar, and the left mandibular third molar (48.72%). Dental eruption was observed more frequently in mandibular teeth (61.28%).

A positive correlation was observed between age and the variables dental wear (r = 0.40), dentin exposure (r = 0.44), dehiscence (r = 0.37), tooth loss (r = 0.28), bone bulging (0.48), dental trauma (r = 0.37), non-functional teeth (r = 0.49), and bone resorption (r = 0.43) (Table 2) using Pearson’s correlation test.

### 3.4. Origins

Of the 180 specimens evaluated, 47 (26.1%) were wildlife animals (in situ), 84 (46.7%) were captive-bred (ex situ), and 49 (27.2%) had no identification regarding their origin. The origin was associated with bone resorption (*p* = 0.0005), fenestration (*p* = 0.05), dehiscence (*p* = 0.03), tooth loss (*p* = 0.02), furcation exposure (*p* = 0.01), and teeth without function (*p* = 0.01), and all lesions were more common in captive-bred animals (Table 3).

Regarding the origin of each species, 12 (50%) specimens of *B. dichotomus* were captive-bred, 1 (4.2%) was a wildlife animal, and 11 (45.8%) lacked this information. On the other hand, *M. americana* exhibited a distribution of 24 (51%) captive-bred, 10 (21.3%) were wildlife animals, and 13 (27.7%) without this information. *S. gouazoubira* presented 22 (51.2%) captive-bred, 15 (34.9%) were wildlife animals, and 6 (13.9%) without information. 

*P. nemorivagus* had 10 (76.9%) specimens captive-bred and 3 (23.1%) were wildlife animals. For *O. bezoarticus*, three (33.3%) were captive-bred, and six (66.7%) lacked this information. For *M. bororo*, six (60%) specimens were captive-bred and four (40%) lacked this information. On the other hand, specimens of *M. chunyi*, *M. pandora*, and *M. bricenii* were wildlife animals.

Among the young animals, six (66.7%) were captive-bred, two (22.2%) were wildlife animals, and one (11.1%) had no available information. As for adult animals, 51 (83.6%) were captive-bred, 7 (11.5%) were wildlife animals, and 3 (4.9%) had no recorded information. In relation to animals whose age was estimated using dentition, 14 (14.9%) adults were captive-bred, 28 (29.8%) wildlife animals, 7 (7.4%) young wildlife animals, and 45 (47.9%) did not have this information. Among senior animals, 13 (81.25%) were captive-bred, and 3 (18.75%) were wildlife animals. 

### 3.5. Dental Lesions

#### 3.5.1. Dental Calculus

Of the 180 animals evaluated, 174 (96.7%) showed signs compatible with the presence of some degree of adhered dental calculus. The right mandibular first molar was the tooth with the highest occurrence of calculus (89.08%), and mandibular teeth were more affected than the maxillary (51.28%). Regarding the degree of dental calculus, 24 (13.79%) animals exhibited grade 1 dental calculus, 99 (56.90%) grade 2, and 51 (29.31%) grade 3 (Figure 1). A negative correlation between the presence of dental calculus and tooth eruption (r = −0.20) (Table 2) was verified by Pearson’s correlation test, meaning younger animals in the tooth eruption phase have a lower frequency of dental calculus. 

#### 3.5.2. Tooth Wear and Dentine Exposure

Out of 180 animals evaluated, 128 (71.1%) showed signs of tooth wear. The right maxillary first premolar had the highest occurrence of dental wear (73.44%), and mandibular teeth were more affected than the maxillary (53.75%). Among the 128 animals with tooth wear, 81 (63.28%) had grade 1 wear, 36 (28.13%) grade 2, and 11 (8.59%) grade 3 (Figure 2). Pearson’s correlation test identified a positive correlation between tooth wear and variables dehiscence (r = 0.45) and furcation exposure (r = 0.39) (Table 2). 

Signs of dentin exposure were observed in 109 (60.6%) of all animals evaluated (Figure 2C,D). The right maxillary first premolar and left mandibular second premolar had the highest prevalence of injury (75.23%). The lesion was more frequent in mandibular teeth (52.18%). Pearson’s correlation test identified a positive association between dentin exposure and the variables dehiscence (r = 0.42), bulging (r = 0.24), and furcation exposure (r = 0.35) and a negative association between dentin exposure and tooth eruption (r = −0.22) (Table 2). 

#### 3.5.3. Tooth Loss

Signs of tooth loss were observed in 43 (23.9%) evaluated animals (Figure 3A). The right maxillary first molar and left maxillary first premolar had the highest rate of tooth loss (30.23%). Tooth loss was greater in maxillary teeth (56.80%). Pearson’s correlation test identified a positive association between tooth loss and the variables furcation exposure (r = 0.25) and non-functional teeth (r = 0.58) (Table 1). No significant difference was noted between tooth loss and the evaluated species. Among the young animals, one (11.1%) captive-bred and one (11.1%) wildlife deer exhibited tooth loss. In adults, 12 (19.7%) captive-bred, 2 (3.3%) wildlife animals, and 1 (1.6%) without data were affected. Among the seniors, nine (56.2%) captive-bred and one (6.3%) wildlife deer exhibited tooth loss.

#### 3.5.4. Non-Functional Teeth 

Of the total number of animals, 38 (21.1%) showed signs of non-functional teeth (Figure 3B). The teeth with the highest occurrence of loss of dental support were the right maxillary second molar and left maxillary second molar (34.21%). The presence of non-functional teeth was more frequent in maxillary teeth (59.46%). Pearson’s correlation denoted a positive association between non-functional teeth and the variables bone resorption (r = 0.29) and tooth eruption (r = −0.24) (Table 2). No significant difference was shown between non-functional teeth and the evaluated species. Among the young animals, one (11.1%) wildlife animal exhibited this condition. In adults, 15 (24.6%) captive-bred and 2 (3.3%) wildlife animals were affected. Among the seniors, 10 (62.5%) captive-bred and 2 (12.5%) wildlife animals exhibited non-functional teeth.

#### 3.5.5. Tooth Fracture

Signs of tooth fracture were observed in 79 (43.9%) evaluated animals (Figure 3C). The teeth with the highest occurrence of fractures were the right maxillary first premolar, left maxillary first molar, and left maxillary second molar (22.78%). Fracture was more common in maxillary teeth (53.82%). Pearson’s correlation test highlighted a positive correlation between furcation exposure (r = 0.26) and bone resorption (r = 0.31) (Table 2). 

#### 3.5.6. Furcation Exposure

Of the total number of animals, 75 (41.7%) showed signs of furcation exposure (Figure 4A). The occurrence of this lesion was more common in the left maxillary third premolar (32%) and maxillary teeth (64.01%) in general. Pearson’s correlation test showed a positive association between furcation exposure and the variables non-functional teeth (r = 0.39) and bone resorption (r = 0.35) (Table 2). In young animals, two (22.2%) captive-bred, one (11.1%) wildlife animal, and one (11.1%) without data exhibited furcation exposure. Among adults, 26 (42.6%) captive-bred and 1 (1.6%) wildlife animal were affected. For the seniors, 10 (62.5%) captive-bred and 2 (12.5%) wildlife animals exhibited furcation exposure.

### 3.6. Bone Lesions

#### 3.6.1. Fenestration

Out of the total number of animals evaluated, 84 (46.7%) showed signs of fenestration (Figure 4B). The maxillary left second premolar and maxillary left first molar were the most affected teeth (26.19%). Maxillary teeth had a higher occurrence of the lesion (65.32%) compared to mandibular teeth. Pearson’s correlation test highlighted that the variables associated with fenestration were dehiscence (r = 0.30), bone bulging (r = 0.26), and bone resorption (r = 0.28) (Table 2). Among the young animals, four (44.4%) captive-bred and one (11.1%) wildlife animal exhibited this condition. In adults, 22 (36.1%) captive-bred, 5 (8.2%) wildlife animals, and 1 (1.6%) without data exhibited it. Among the seniors, 10 (62.5%) captive-bred and 1 (6.3%) wildlife animal exhibited fenestration.

#### 3.6.2. Dehiscence

Among the evaluated animals, 96 (53.3%) showed signs of dehiscence (Figure 4B). The teeth with the highest occurrence of the lesion were the right maxillary first molar, right maxillary third premolar, left maxillary third premolar, and left maxillary first molar (35.42%). There was a greater occurrence of dehiscence in maxillary teeth (62.45%). Pearson’s correlation test denoted a significant correlation between dehiscence and the variables bulging (r = 0.30), furcation exposure (r = 0.38), and bone resorption (r = 0.45) (Table 2). In young animals, two (22.2%) captive-bred and one (11.1%) without data exhibited dehiscence. Among adults, 31 (50.8%) captive-bred, 2 (3.3%) wildlife animals, and 1 (1.6%) without data were affected. For the seniors, 10 (62.5%) captive-bred and 3 (18.8%) wildlife animals exhibited this condition.

#### 3.6.3. Bone Bulging

Regarding bone bulging, 31 (17.2%) animals showed compatible signs (Figure 4C). The left mandibular second molar had the highest occurrence of bulging (38.71%), and the mandibular bone was the most affected by the lesion (84.27%). Positive correlations were identified with furcation exposure (r = 0.44), non-functional teeth (r = 0.35), and bone resorption (r = 0.31) (Table 2).

#### 3.6.4. Bone Resorption 

Of the total number of animals, 84 (46.7%) showed signs of bone resorption (Figure 4D). In 33 (36.59%) animals, the first left incisor was the tooth most affected by bone resorption. Moreover, bone resorption was more common in mandibular teeth (53.72%). There was a negative correlation with the tooth eruption (r = −0.23). In young animals, two (22.2%) captive-bred exhibited bone resorption. Among adults, 27 (44.3%) captive-bred, 3 (4.9%) wildlife animals, and 2 (3.3%) without data were affected. For the seniors, 12 (75%) captive-bred and 3 (18.8%) wildlife animals exhibited bone resorption.

## 4. Discussion

This study revealed a high prevalence of bone and dental alterations in dry skulls of neotropical deer species. Every analyzed species had at least one specimen with some alteration, and species with fewer available specimens had a lower frequency of lesions.

The age of the evaluated individuals positively correlated with most analyzed bone and dental alterations, notably dental wear, dentin exposure, bone bulging, furcation exposure, non-functional teeth, and bone resorption. In a retrospective study that evaluated an ex situ population of *B. dichotomus* from the Marsh Deer Conservation Center, the place of origin of some skulls in the present study, it was found that the probability of developing oral diseases increased eight times with each change in age [25].

Studies from North America and Europe also associated age with conditions compromising the oral integrity of animals [21,22,23]. This probably occurs due to cumulative exposure throughout the animal’s life to modifying factors that contribute to the development of oral changes [16]. The current study suggests that advancing age may be a crucial factor in the occurrence and progression of oral alterations in deer.

Besides age, animal origin was another factor associated with a higher prevalence of lesions. All lesions were more frequent in captive-bred animals; a finding supported by a previous study [19]. Our data indicate that captive-bred animals had significantly higher instances of dental calculus, tooth wear, fenestration, dehiscence, tooth loss, furcation exposure, non-functional teeth, and bone resorption compared to wildlife animals. In a study carried out in Spain, a possible association between diet and the development of dental lesions was observed. Animals that consumed more sprouts and fruits had a higher prevalence of dental lesion [23]. 

The diet in the wild is much more varied than in captivity. For example, the species *O. bezoarticus*, when in its natural habitat, can ingest more than 55 different native plant species, including more than 74 parts of these plants, such as shoots, leaves, flowers, flower buds, fruits, and seeds of these plants [3]. It is likely that the limitation in the variety of food of animals kept under human care, when compared to those in the wild, allows for greater exposure to modifying factors associated with the etiopathogenesis of periodontal disease. Furthermore, it is observed that animals kept under human care in captivity present a longer life expectancy. This trend may be associated with a higher incidence of bone and dental alterations in animals kept in captivity, as identified in the study [28].

In the present study, no variable evaluated showed significant dependence on sex. In a previous study with the species *B. dichotomus*, no significant association was also observed between the occurrence of oral diseases and the sex variable [25]. In Spain, it was possible to observe that the prevalence of lesions between *Cervus elaphus* males and *Dama dama* females was higher than in the opposite sex, probably due to the difference in feeding behavior between males and females of each species [23]. This observation supports the theory that diet may play a role in predisposing the appearance of these changes.

In the present study, a high prevalence of tooth wear (71.1%) was observed in the animals evaluated. Evidence suggests that cervids with high rates of tooth wear experience reduced survival, reproductive performance, mass, and body condition [11,12,13]. Tooth wear has been mainly associated with dentin exposure and is considered a pathological process when this occurs [29,30]. Furthermore, no significant differences were found in the prevalence of wear between the mandibular and maxillary teeth of the animals evaluated, which shows that tooth wear occurs in all groups of teeth.

The present study observed a low association between tooth wear and some periodontal disease indicators, such as dehiscence and furcation exposure. Although the two diseases have different etiologies, it is common for them to occur simultaneously, as these changes may share some factors such as age and the presence of some bacteria, such as those belonging to the *Prevotella* genus [31,32]. These bacteria degrade salivary proteins and, possibly, the acquired salivary film, affecting tooth protection against erosive actions that cause tooth wear [32]. In a retrospective study of wild ruminants carried out in Switzerland, excessive/irregular tooth wear or deformation was identified as the cause of death in 70% of notifications [33]. Tooth wear has also been described in other wild animals, such as elks (*Alces alces*), lemurs (*Lemur catta*), and giraffes (*Giraffa camelopardalis*) [21,34,35]. 

In a recent study, Saraiva et al. [36] demonstrated that the pigmented deposits that form on the dental crown of cattle represent a true bacterial biofilm resulting from the metabolism of microorganisms and associated with bovine periodontitis. In the present study, the occurrence of dental calculus was identified in almost all the skulls evaluated at some level of classification (96.7%). More than 80% of the animals had a dental calculus score equal to or greater than two, indicating that a large proportion of the animals studied were predisposed to developing periodontal disease, if they had not developed it. Furthermore, there is evidence that the degree of dental calculus is associated with the intensity of gingival recession and may, therefore, be related to the occurrence of severe periodontitis [31,37]. The most affected tooth was the right mandibular first molar (89.08%). Molars are commonly most affected, as they are close to the opening of the salivary duct and saliva is the main source of minerals for the formation of dental calculus through biofilm mineralization [38]. 

Changes related to bone resorption, fenestration, and dehiscence were also common in the present study (46.7%, 46.7%, 53.3%, respectively). These three conditions indicate the presence of periodontal disease in the animal [39,40,41,42]. Bone resorption has already been described in several species of cervids in different countries, such as Brazil, Canada, the United States of America, and South Georgia Island [21,22,25,39]. 

Furcation exposure has already been reported in *B. dichotomus* and *C. elaphus* L. [20,25]. The occurrence of furcation exposure is present in the advanced stages of periodontal disease [43,44]. In the present study, a significant prevalence of furcation exposure (41.7%) was observed among the evaluated animals, indicating a high prevalence of periodontitis. This finding was associated with non-functional teeth, as both conditions occur when there is a loss of supporting tissues [16].

Facial bulging due to dental abscesses was commonly described in deer studies and was the most frequent lesion in the clinical records of *B. dichotomus* (65.3%), likely because it is easily identifiable compared to other lesions requiring detailed clinical examination for detection [25]. This bulging was also reported in *R. tarandus greenlandicus* specimens in Canada and in the USA [17,18]. Bone bulging of the mandible and maxilla presented a high prevalence in the present study (17.2%). Bone bulging was associated with dentin exposure, probably because dentin exposure can result in exposure of the pulp cavity, which predisposes the entry of microorganisms, leading to the formation of odontogenic abscesses [45,46]. 

Severe periodontitis can lead to tooth loss due to severe bone destruction and loss of clinical tooth attachment [16]. Both tooth loss (23.9%) and non-functional teeth (21.1%) were lesions frequently found in the animals in the present study, as has already been reported in studies on cervids of the species *R. tarandus* [39]. In Brazil, a prevalence of 22.4% of tooth loss was identified in *B. dichotomus* [25]. In Alaska, a prevalence of 33.7% was identified in *R. tarandus* [18]. Tooth fracture also had a high incidence (43.9%) and has already been reported in other studies with different species of deer [12,17,18,20,21,22,25,39]. 

Despite the high prevalence of bone and dental alterations observed, only 11 animals had documented dental issues while alive. This discrepancy may be due to the challenges of diagnosing periodontal diseases in living ruminants, as it requires a detailed inspection of the oral cavity, which is made difficult by the anatomy of these animals [16]. The lack of complete clinical records for all animals examined restricted the ability to correlate cranial bone and dental changes with clinical history and risk factors. This emphasizes the necessity for clinical and complementary oral examinations and expanding scientific knowledge regarding the oral health of these animals, as well to care with the diet offered to these animals. This information is vital for the creation of management and conservation strategies aimed at preserving these species and ensuring their health and welfare. It is noteworthy that out of the 11 species analyzed, *B. dichotomus*, *M. bororo*, *M. nana*, *H. antisensis*, *M. rufina*, *M. bricenii*, *M. pandora*, and *H. bisulcus* are classified as vulnerable and one is endangered [7]. Oral affections can significantly impact the conservation of these species, as these conditions can jeopardize reproduction and population viability, especially in endangered species [11,12,13]. 

## 5. Conclusions

The study highlights that all eleven species assessed were prone to developing bone and dental lesions. Notably, older animals, particularly those in captivity, displayed a higher frequency of such lesions. This underscores the association between age and captivity as risk factors for the occurrence of these alterations in cervids. Therefore, prioritizing the monitoring of oral health and diet, especially in captive settings, is crucial for the conservation endeavors aimed at safeguarding these species.

## Figures and Tables

**Figure 1 animals-14-01892-f001:**
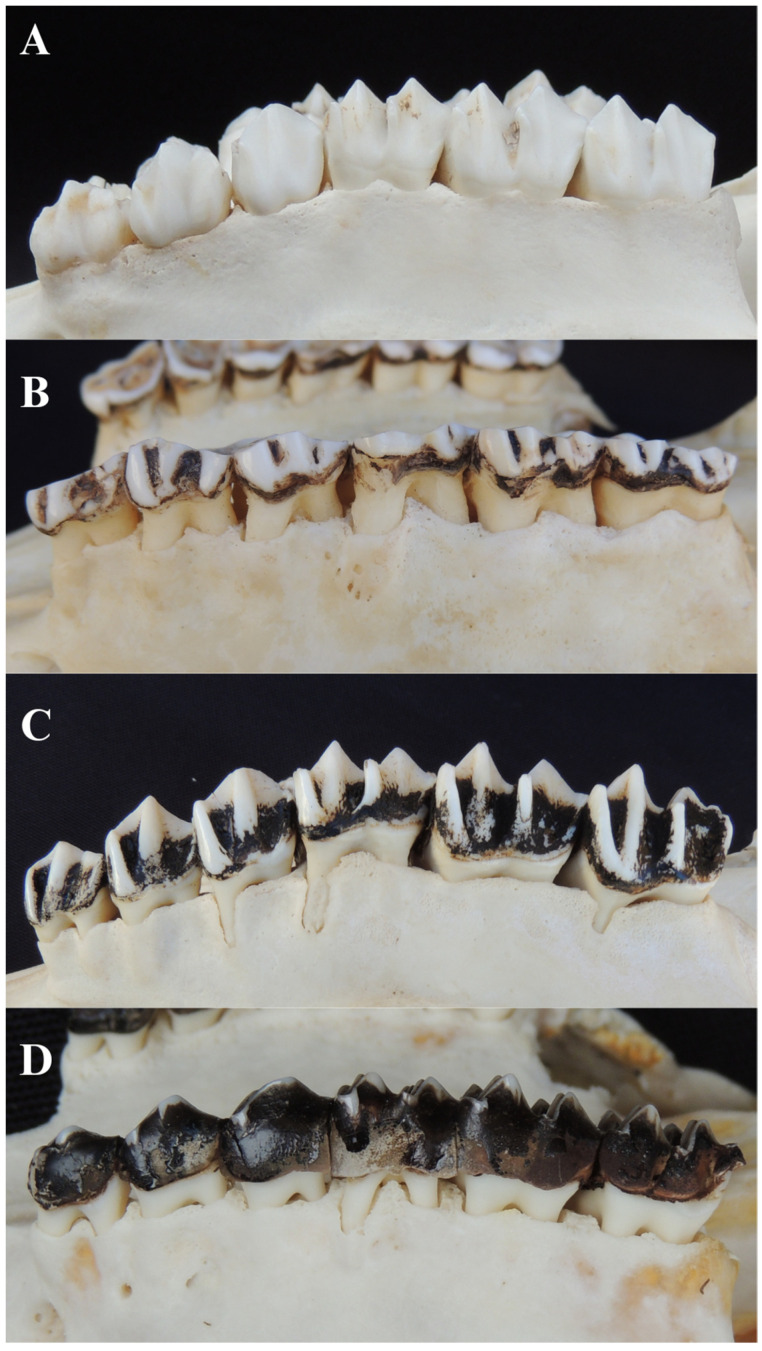
Different degrees of dental calculus in right maxillary premolars and molars. (**A**) Grade 0 dental calculus in an eight-year-old female *Mazama americana*. (**B**) Grade 1 dental calculus in a female *M. americana*. (**C**) Grade 2 dental calculus in a female *Blastocerus dichotomus*. (**D**) Grade 3 dental calculus in a female *M. chunyi*.

**Figure 2 animals-14-01892-f002:**
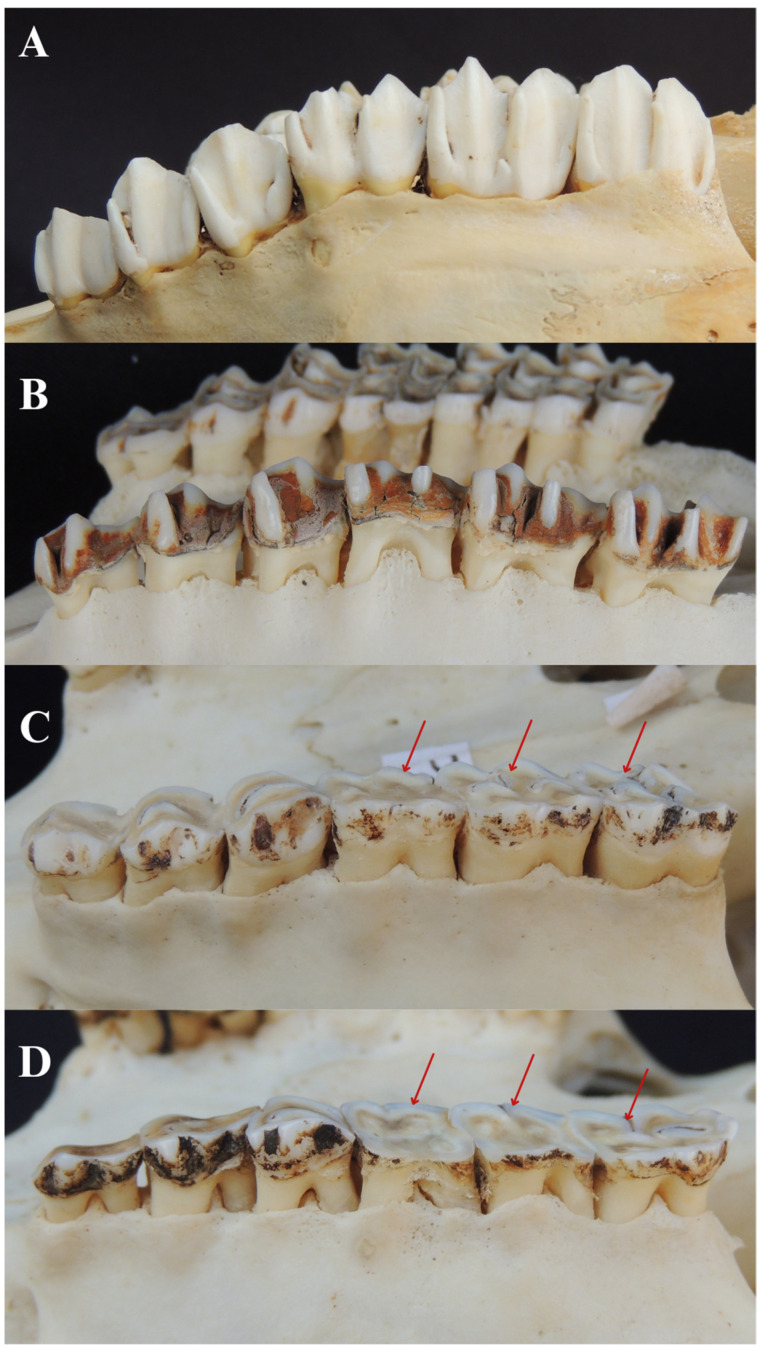
Different degrees of tooth wear in right maxillary premolars and molars. (**A**) Grade 0 tooth wear in a male *Ozotoceros bezoarticus*. (**B**) Grade 1 tooth wear in a ten-year-old female *B. dichotomus*. (**C**) Grade 2 tooth wear in a six-year-old male *M. americana*. Red arrow indicating dentin exposure. (**D**) Grade 3 tooth wear on molar teeth of a male *Passalites nemorivagus*. Red arrow indicating dentin exposure.

**Figure 3 animals-14-01892-f003:**
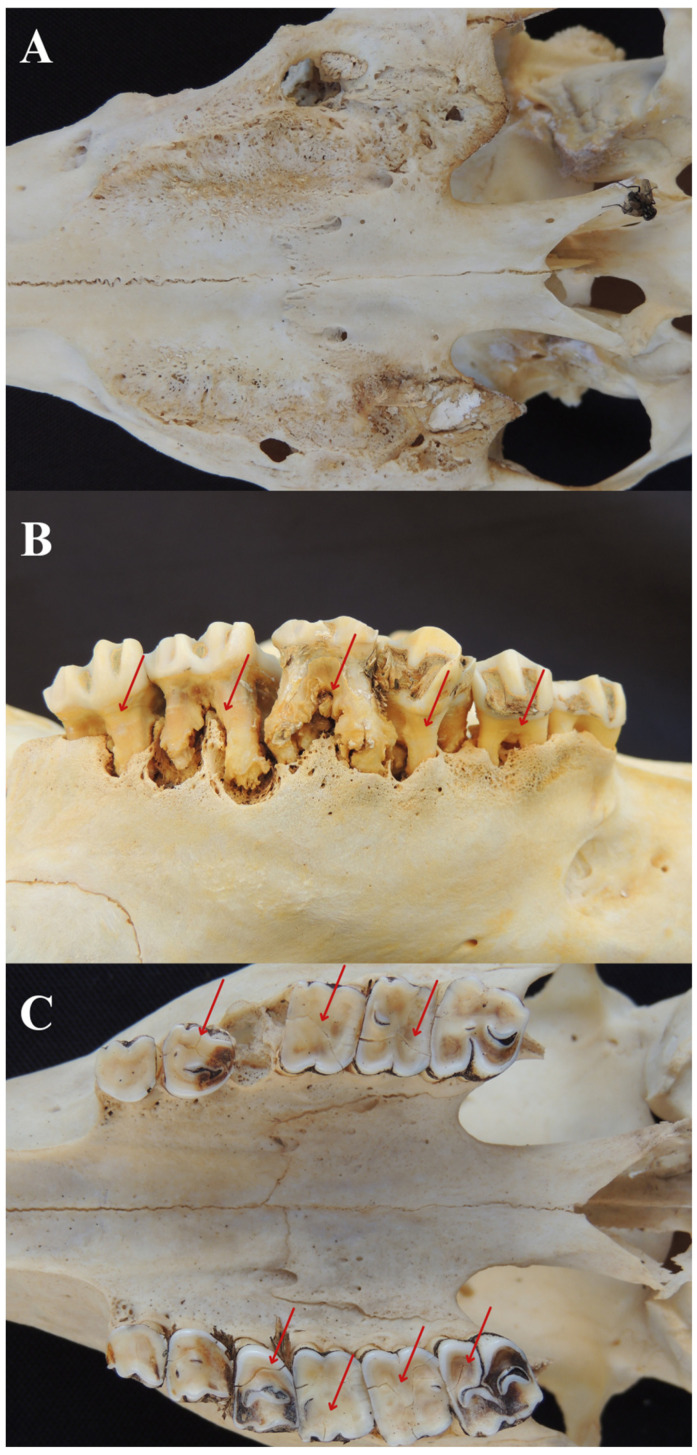
(**A**) Absence of all maxillary teeth in a 15-year-old male *Mazama americana*. (**B**) Non-functional left maxillary second and third premolar teeth and all molars highlighted by the red arrow of a ten-year-old female *Blastocerus dichotomus*. (**C**) Presence of tooth fracture in a female *Passalites nemorivagus* in the maxillary teeth: second premolar, first and second left molars, third premolar, and right molars highlighted by the red arrow.

**Figure 4 animals-14-01892-f004:**
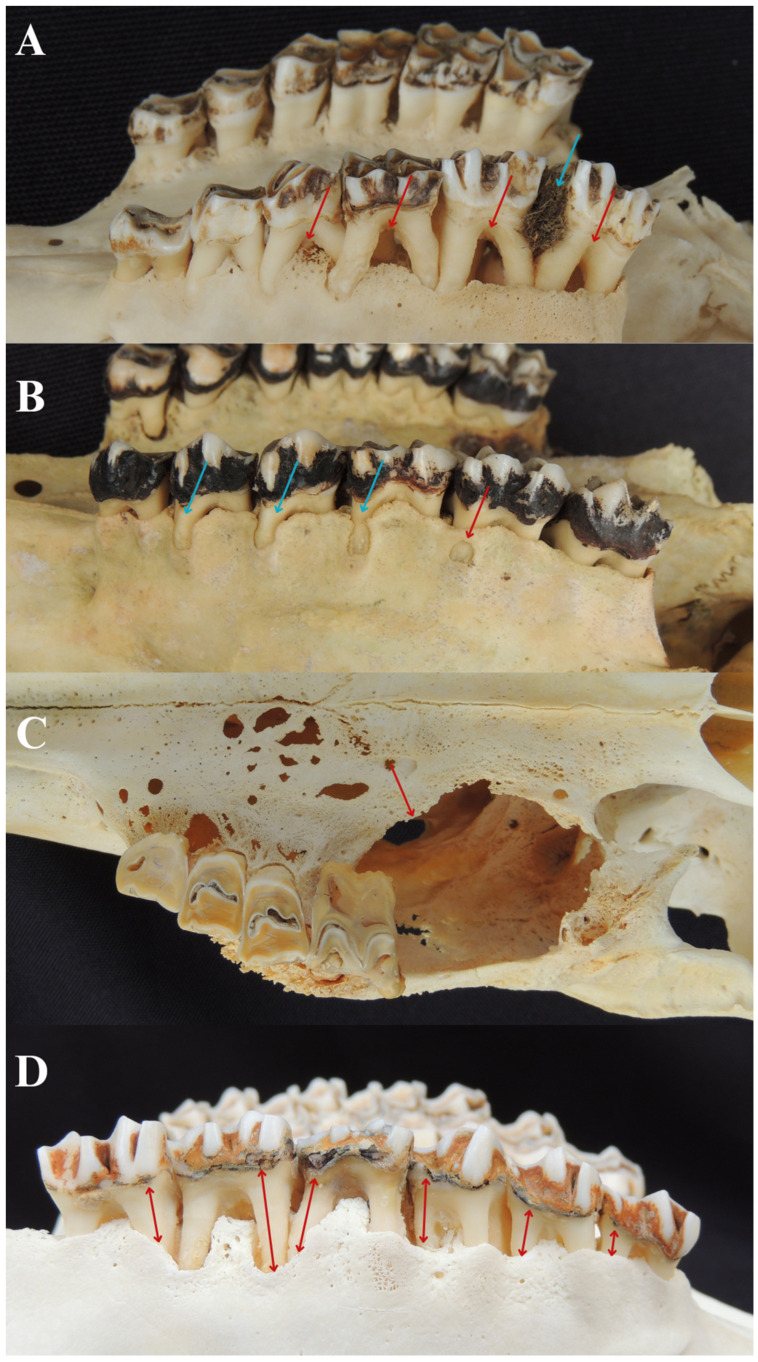
(**A**) Furcation exposure evidenced by the red arrow on the third premolar and right molars of the maxilla in a ten-year-old male *M. nana* and food impaction between the second and third molar teeth evidenced by the blue arrow. (**B**) Presence of fenestration in the maxillary bone adjacent to the right second molar evidenced by the red arrow and dehiscences in the right maxillary bone adjacent to the second and third premolar and first molar evidenced by blue arrows in a female *M. bororo*. (**C**) Presence of bone bulging evidenced by a red arrow in a 14-year-old female *B. dichotomus*. (**D**) Presence of bone reabsorption in the cementoenamel junction and the alveolar crest on the buccal surface of the left maxillary teeth in a ten-year-old female *B. dichotomus* highlighted by the red arrow.

**Table 1 animals-14-01892-t001:** Absolute and relative frequencies of dental and bone lesions in specimens of the species *Blastocerus dichotomus*, *Ozotoceros bezoarticus*, *Mazama bororo*, *Subulo gouazoubira*, *Passalites nemorivagus*, *Mazama americana*, *Mazama temama*, *Mazama pandora*, *Mazama bricenii*, *Mazama chunyi*, *Mazama nana*, and *Mazama* sp.

Species		Frequencies of Dental and Bone Alterations																					
		Dental Calculus		Tooth Wear		Dentine Exposure		Fenestration		Dehiscence		Tooth Loss		Bone Bulging		Tooth Fracture		Furcation Exposure		Non-Functional Teeth		Bone Resorption	
Frequency	Total	n	%	n	%	n	%	n	%	n	%	n	%	n	%	n	%	n	%	n	%	n	%
*Blastocerus dichotomus*	*24*	23	95.8	19	79.2	15	62.5	17	70.8	20	83.3	10	41.6	10	41.6	19	79.2	12	50	8	33.3	20	83.3
*Ozotoceros bezoarticus*	*9*	6	66.6	4	44.4	3	33.3	6	66.6	4	44.4	1	11.1	1	11.1	4	44.4	4	44.4	0	0	5	55.5
*Mazama bororo*	*10*	10	100	7	70	5	50	4	40	3	30	2	20	0	0	3	30	3	30	0	0	3	30
*Subulo gouazoubira*	*43*	43	100	31	72.1	27	62.8	20	46.5	23	53.5	7	16.3	2	4.6	14	32.5	13	30.2	10	23.2	16	37.2
*Passalites nemorivagus*	*13*	13	100	10	76.9	8	61.5	6	46.1	9	69.2	4	30.8	3	23.0	5	38.5	12	92.3	4	30.8	4	30.8
*Mazama americana*	*47*	47	100	34	72.3	29	61.7	18	38.3	24	51.1	12	25.5	9	19.1	24	51.0	23	48.9	13	27.6	27	57.4
*Mazama temama*	*4*	4	100	3	75	3	75	2	50	2	50	0	0	0	0	2	50	0	0	0	0	2	50
*Mazama pandora*	*1*	1	100	1	100	1	100	0	0	0	0	0	0	0	0	0	0	1	100	0	0	0	0
*Mazama bricenii*	*1*	1	100	1	100	1	100	1	100	0	0	0	0	0	0	1	100	0	0	0	0	0	0
*Mazama chunyi*	*2*	2	100	1	50	1	50	0	0	1	50	0	0	0	0	0	0	1	50	0	0	0	0
*Mazama nana*	4	4	100	3	75	3	75	1	25	2	50	3	75	2	50	1	25	2	50	3	75	2	50
*Mazama* sp.	*12*	11	91.7	6	50	6	50	8	66.7	4	33.3	1	8.3	4	33.3	2	16.7	2	16.7	0	0	2	16.7

**Table 2 animals-14-01892-t002:** Pearson’s simple correction coefficients between the characters, respectively, age and bone and dental lesions in neotropical deer.

Parameters *	Correction Coefficients												
	AG *	DC	TW	DE	FE	DS	TL	BB	TF	EF	TE	NFT	BR
AG		0.10	0.40 ^a^	0.44 ^a^	0.09	0.37 ^a^	0.28 ^a^	0.48 ^a^	0.37 ^a^	−0.46 ^a^	−0.45 ^a^	0.49 ^a^	0.43 ^a^
DC			0.10	0.10	−0.10	0.00	0.00	0.00	0.10	0.10	−0.20 ^a^	−0.20	0.00
TW				0.92 ^a^	0.04	0.45 ^a^	0.20	0.21	0.10	0.39 ^a^	−0.19	0.17	0.16
DE					0.02	0.42 ^a^	0.13	0.24 ^a^	0.12	0.35 ^a^	−0.22 ^a^	0.16	0.10
FE						0.30 ^a^	−0.07	0.26 ^a^	0.19	0.00	0.15	0.03	0.28 ^a^
DS							0.14	0.30 ^a^	0.01	0.38 ^a^	−0.17	0.18	0.45 ^a^
TL								0.09	−0.07	0.25 ^a^	−0.15	0.58 ^a^	0.10
BB									0.19	0.44 ^a^	−0.23 ^a^	0.35 ^a^	0.31 ^a^
TF										0.26 ^a^	−0.08	0.13	0.31 ^a^
EF											−0.29 ^a^	0.39 ^a^	0.35 ^a^
TE												−0.24	−0.23 ^a^
NFT													0.29 ^a^

^a^ Significant to 5% probability; * AG = age; DC = dental calculus; TW = tooth wear; DE = dentin exposure; FE = fenestration; DS = dehiscence; TL = tooth loss; BB = bone bulging; TF = tooth fracture; FE = furcation exposure; TE = tooth eruption; NFT = non-functional teeth; and BR = bone resorption.

**Table 3 animals-14-01892-t003:** Absolute frequency and relative frequency of bone and dental lesions in skulls from free-living and captive specimens of neotropical deer.

	Captive-Bred Deer		Free-Living Deer		Not Identified		Total	
Dental and Bone Alterations	n	%	n	%	n	%	n	%
Dental calculus grade 1	11	45.8	3	12.5	10	41.7	24	100
Dental calculus grade 2	49	49.5	28	28.3	22	22.2	99	100
Dental calculus grade 3	23	45.1	16	31.4	12	23.5	51	100
Tooth wear grade 1	33	40.7	23	28.4	25	30.9	81	100
Tooth wear grade 2	19	52.8	8	22.2	9	25	36	100
Tooth wear grade 3	10	90.9	1	9.1	0	0	11	100
Dentin exposure	57	52.3	28	25.7	24	22	109	100
Fenestrations	43	51.2	16	19	25	29.8	84	100
Dehiscence	52	54.2	20	20.8	24	25	96	100
Tooth loss	25	58.1	6	14	12	27.9	43	100
Bone bulging	20	64.5	6	19.4	5	16.1	31	100
Tooth fracture	34	43	23	29.1	22	27.9	79	100
Furcation exposure	46	61.3	15	20	14	18.7	75	100
Non-functional teeth	29	76.3	7	18.4	2	5.3	38	100
Bone resorption	48	57.1	12	14.3	24	28.6	84	100

## Data Availability

The raw data supporting the conclusions of this article will be made available by the author on request.

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
