# Peer review of "Prevalence and Risk Factors of Bone and Dental Lesions in Neotropical Deer"

_animals, 2024, doi:10.3390/ani14131892_

Round 1

Reviewer 1 Report

Comments and Suggestions for Authors

I believe you have conducted an extensive research effort with a considerable sample size, based on a series of osteological remains of a museum nature. Therefore, I understand that these are bone remains that are optimally preserved for study. I suggest you provide this information to ensure there is no doubt regarding the value of studying these remains

On the other hand, there is a deficiency in Table 1, which I understand may be related to the uploading process. Capital letters have been displaced, making it difficult to follow. I kindly request that you consider improving it

 conclusions are clear, and the results are as expected considering your exposition. However, as a personal query: Do you think it would be possible to reduce all these injuries sufficiently to consider it an improvement in the conservation of certain species?

Author Response

Dear Chief Editor and reviewer,

We appreciate the suggestions and corrections made in our manuscript. All placements were extremely relevant. Corrections were made in the text and are highlighted in yellow. Below, we describe the comments and responses to them. We hope to meet the requests made. Any question we are available.

# Reviewer 1

1) I believe you have conducted an extensive research effort with a considerable sample size, based on a series of osteological remains of a museum nature. Therefore, I understand that these are bone remains that are optimally preserved for study. I suggest you provide this information to ensure there is no doubt regarding the value of studying these remains.

Corrected. This information was inserted in the text (Lines 91-92).

2) On the other hand, there is a deficiency in Table 1, which I understand may be related to the uploading process. Capital letters have been displaced, making it difficult to follow. I kindly request that you consider improving it.

Corrected. Table 1 was reformulated for a better understanding (Lines 249-252).

3) Conclusions are clear, and the results are as expected considering your exposition. However, as a personal query: Do you think it would be possible to reduce all these injuries sufficiently to consider it an improvement in the conservation of certain species?

While the high prevalence of bone and dental alterations underscores the challenges of diagnosing and managing these conditions in cervids, it is indeed possible to reduce these injuries and improve conservation efforts, especially for animals kept in captivity, by implementing regular oral health monitoring and studying the diets involved to identify those associated with a high frequency of lesions. This proactive approach can prevent animals from reaching critical levels of dental and bone health issues, thereby significantly enhancing the conservation of these vulnerable and endangered species.

Reviewer 2 Report

Comments and Suggestions for Authors

Abstract

-              Is the abstract structured or unstructured?

-              Why only results were separated while the background, methods, and conclusions were not indicated?

-              A high prevalence of bone and dental lesions was observed in all analyzed species. Kindly state the actual value and provide the 95% CI

-              “Animal age positively 41 correlated with most bone and dental alterations, indicating that older animals showed more lesions. Additionally, the prevalence of these alterations was similar between sexes. Moreover, all lesions were more common in captive-bred animals, likely attributed to their older age and a less diverse diet” The information should be supported with the actual values and 95% CI, rather than merely stating high or low prevalence. The correlation coefficient should be provided as well. 

Introduction

= “Given that eight out of the eleven species in the study are classified as threatened or vulnerable” the 11 species were not stated anywhere in the previous paragraphs

-              The justification and rationale for conducting this study should be improved with more information and proper citations. 

-              Overall, the introduction needs to be revamped because there appears to be too much information hidden in each paragraph as reflected in the citations [2-8], [17-25]. I suggest elaborating on the information provided in these citations so that the reader can gain a better understanding of the information gathered from each cited article

-              Another important point is the study method used, which entails visual observation of deer skulls, rather than sampling live deer as such will be very challenging to conduct in the field. It should be stated somewhere in the introduction why this methodology was employed – which may reflect the author’s attempts to bridge the research gap. 

-               

Methods

-              Sampling: these deer species were not stated in the introduction. I suggest providing information from the literature on these species and why they were selected for this study – probably due to their vulnerability and possible extinction….

-              The specific type of sampling method should be mentioned. 

-              Feeding management of deer under human care: the source of the information should be provided because it is currently unclear where the authors obtained the data

-              Data analysis section: why was the correlation between the variables conducted and how is it different from the analysis performed using the Chi-square test? For better understanding, the outcome measures and independent variables should be clearly stated in the section to clearly depict the correlation test and risk factor analysis.

-              Why was the binary logistic regression model not considered in the analysis? Was there any violation of the assumption for the Chi-square test (i.e., some cells having less than 5)?

Results

-              Please provide the specific data and figures, and do not limit them to p-values 

-              All prevalence estimates should be supported by the 95% CI

-              For the “Sex” category – were the conditions to use the Chi-square test violated? Maybe the Fischer exact test can be reported instead

-              Correlation should be clearly stated either as positive or negative for clearer interpretation. 

-              Table 1: species names should be italicized in the caption

Discussion

-              I will comment on the discussion once the earlier comments are addressed

-              However, the authors seem not to highlight the limitations of the study in the article. 

Comments on the Quality of English Language

Minimal revision required

Author Response

Dear Chief Editor and reviewer,

We appreciate the suggestions and corrections made in our manuscript. All placements were extremely relevant. Corrections were made in the text and are highlighted in yellow. Below, we describe the comments and responses to them. We hope to meet the requests made. Any question we are available.

# Reviewer 2

Abstract

1) Is the abstract structured or unstructured? Why only results were separated while the background, methods, and conclusions were not indicated?

Corrected. Abstract should not be structured and was corrected.

2) “A high prevalence of bone and dental lesions was observed in all analyzed species. Kindly state the actual value and provide the 95% CI.

We appreciate the suggestion, but since the abstract only allows 200 words, it is not possible to include the results of all the evaluated lesions. Therefore, if the reviewer does not mind, the results were described in the results section and only the most important results were kept in the abstract.

3) “Animal age positively correlated with most bone and dental alterations, indicating that older animals showed more lesions. Additionally, the prevalence of these alterations was similar between sexes. Moreover, all lesions were more common in captive-bred animals, likely attributed to their older age and a less diverse diet” The information should be supported with the actual values and 95% CI, rather than merely stating high or low prevalence. The correlation coefficient should be provided as well. 

We appreciate the suggestion, but since the abstract only allows 200 words, it is not possible to include the results of all the evaluated lesions. Therefore, if the reviewer does not mind, the results were described in the results section and only the most important results were kept in the abstract.

Introduction

3) “Given that eight out of the eleven species in the study are classified as threatened or vulnerable” the 11 species were not stated anywhere in the previous paragraphs.

Corrected. This information was corrected in the text for a better understanding (lines 80-81).

 4) The justification and rationale for conducting this study should be improved with more information and proper citations. 

Corrected. More information about the justification and rationale for conducting the study was inserted in the text (lines 71-88).

5) Overall, the introduction needs to be revamped because there appears to be too much information hidden in each paragraph as reflected in the citations [2-8], [17-25]. I suggest elaborating on the information provided in these citations so that the reader can gain a better understanding of the information gathered from each cited article.

Corrected. The information provided in the citations was added to the text (lines 57-59, 71-73).

6) Another important point is the study method used, which entails visual observation of deer skulls, rather than sampling live deer as such will be very challenging to conduct in the field. It should be stated somewhere in the introduction why this methodology was employed – which may reflect the author’s attempts to bridge the research gap. 

Corrected. This information was added to the text (lines 84-88).

Methods

7) Sampling: these deer species were not stated in the introduction. I suggest providing information from the literature on these species and why they were selected for this study – probably due to their vulnerability and possible extinction.

Corrected. This information was added to the text (lines 114-120).

8) The specific type of sampling method should be mentioned. 

Corrected. This information was added to the text (lines 120-121).

9) Feeding management of deer under human care: the source of the information should be provided because it is currently unclear where the authors obtained the data.

Corrected. This information was added to the text (line 147).

10) Data analysis section: why was the correlation between the variables conducted and how is it different from the analysis performed using the Chi-square test? For better understanding, the outcome measures and independent variables should be clearly stated in the section to clearly depict the correlation test and risk factor analysis.

The correlation between variables was conducted to understand the strength and direction of the relationship between two continuous variables. This analysis helps to identify whether an increase or decrease in one variable is associated with an increase or decrease in another variable. While correlation analysis is used for continuous variables, the Chi-square test is employed to examine the relationship between categorical variables. Specifically, the Chi-square test assesses whether there is a significant association between two categorical variables by comparing the observed frequencies in each category to the expected frequencies if there were no association. Corrected, the outcome measures and independent variables were stated for better understanding (Lines 170-181).

11) Why was the binary logistic regression model not considered in the analysis? Was there any violation of the assumption for the Chi-square test (i.e., some cells having less than 5)?

The binary logistic regression model was not considered due to the sample size constraints. Additionally, the assumptions for the Chi-square test were carefully checked, ensuring that all cells had expected frequencies of at least 5, thus validating the use of the Chi-square test in this analysis. If violations were found, alternative methods such as Fisher's exact test were analysed and compared with Chi-square result. This information was included in the text (Lines 165-166).

Results

12) Please provide the specific data and figures, and do not limit them to p-values.

Corrected. Specific data and figures were displayed in Table 1 but for a better understanding, some information was inserted in the text.

13) All prevalence estimates should be supported by the 95% CI

Unfortunately, we do not have this data. But if the reviewer finds it relevant, we will request it.

14) For the “Sex” category – were the conditions to use the Chi-square test violated? Maybe the Fischer exact test can be reported instead.

Corrected. Results of Fisher test were reported for the “sex” category (lines 184-185).

15) Correlation should be clearly stated either as positive or negative for clearer interpretation. 

Corrected throughout the text. It was mentioned when the correlation was positive or negative.  

16) Table 1: species names should be italicized in the caption.

Corrected. Lines 248-251

Discussion

17) I will comment on the discussion once the earlier comments are addressed. However, the authors seem not to highlight the limitations of the study in the article. 

Some limitations of the study were included in the discussion (Lines 521-523).

Reviewer 3 Report

Comments and Suggestions for Authors

The topic of the manuscript is very interesting and brings substantial results. The study is well designed and processed, it contains a sufficient number of samples. Here are a few notes:

·         “However, lesions were more common in captive-bred animals, likely due to their older age and a less diverse diet.“ – I think you confirm this assumption statistically. Please, add the comparison of captive-bred and wild animals in the same age group.

·         The information in which years the samples were obtained is missing.

·         Add information regarding the origin of the samples, or (if possible) also more details about the breeding of captive-bred animals.

·         Do you have some data regarding health problems (especially feeding problems) for at least some of the captive-bred animals? Is the cause of death known?

·         Do you know what areas the wild animals were from?

·         Table 1, lines 175-177: Italics is missing.

·         Table 1: What data is shown in the second to ninth columns?

·         Table 2: Consider whether it wouldn't be clearer to mark only significant results.

·         Table 3: Why is a after DC1?

Author Response

Dear Chief Editor and reviewer,

We appreciate the suggestions and corrections made in our manuscript. All placements were extremely relevant. Corrections were made in the text and are highlighted in yellow. Below, we describe the comments and responses to them. We hope to meet the requests made. Any question we are available.

# Reviewer 3

The topic of the manuscript is very interesting and brings substantial results. The study is well designed and processed; it contains a sufficient number of samples. Here are a few notes:

1) “However, lesions were more common in captive-bred animals, likely due to their older age and a less diverse diet.“ – I think you confirm this assumption statistically. Please, add the comparison of captive-bred and wild animals in the same age group.

Corrected. Results about comparison of captive-bred and wild animals in the same age group were included in the text as suggested (Lines 340-343, 351-354, 372-375, 383-386, 394-397, 408-411).

2) The information in which years the samples were obtained is missing.

Corrected. This information was inserted in the text (lines 98-99).

3) Add information regarding the origin of the samples, or (if possible) also more details about the breeding of captive-bred animals.

Corrected. Information about the origin of the samples was included in the text (lines 114-120).

4)  Do you have some data regarding health problems (especially feeding problems) for at least some of the captive-bred animals? Is the cause of death known?

Corrected. This information was inserted in the text (lines xx-xx).

5) Do you know what areas the wild animals were from?

Corrected. This information was inserted in the text (lines 118-120).

6) Table 1, lines 175-177: Italics is missing.

Corrected.

7) Table 1: What data is shown in the second to ninth columns?

Corrected. Table 1 was reformulated.

8) Table 2: Consider whether it wouldn't be clearer to mark only significant results.

Corrected. Only significant results were marked.

9) Table 3: Why is a after DC1?

Corrected. Table 3 caption was reformulated for a better understanding.

Round 2

Reviewer 2 Report

Comments and Suggestions for Authors

The article has been substantially improved and thanks for considering my comments in the revised version. 

Comments on the Quality of English Language

Minor revision required

Reviewer 3 Report

Comments and Suggestions for Authors

Thank you for making the requested changes. I recommend the manuscript for acceptance.